# Can Active Memory Replace Attention?

**Łukasz Kaiser**
Google Brain
lukaszkaiser@google.com

**Samy Bengio**
Google Brain
bengio@google.com

## Abstract

Several mechanisms to focus attention of a neural network on selected parts of its input or memory have been used successfully in deep learning models in recent years. Attention has improved image classification, image captioning, speech recognition, generative models, and learning algorithmic tasks, but it had probably the largest impact on neural machine translation.

Recently, similar improvements have been obtained using alternative mechanisms that do not focus on a single part of a memory but operate on all of it in parallel, in a uniform way. Such mechanism, which we call *active memory*, improved over attention in algorithmic tasks, image processing, and in generative modelling.

So far, however, active memory has not improved over attention for most natural language processing tasks, in particular for machine translation. We analyze this shortcoming in this paper and propose an extended model of active memory that matches existing attention models on neural machine translation and generalizes better to longer sentences. We investigate this model and explain why previous active memory models did not succeed. Finally, we discuss when active memory brings most benefits and where attention can be a better choice.

## 1  Introduction

Recent successes of deep neural networks have spanned many domains, from computer vision [1] to speech recognition [2] and many other tasks. In particular, sequence-to-sequence recurrent neural networks (RNNs) with long short-term memory (LSTM) cells [3] have proven especially successful at natural language processing (NLP) tasks, including machine translation [4, 5, 6].

The basic sequence-to-sequence architecture for machine translation is composed of an RNN encoder which reads the source sentence one token at a time and transforms it into a fixed-sized state vector. This is followed by an RNN decoder, which generates the target sentence, one token at a time, from the state vector. While a pure sequence-to-sequence recurrent neural network can already obtain good translation results [4, 6], it suffers from the fact that the whole sentence to be translated needs to be encoded into a single fixed-size vector. This clearly manifests itself in the degradation of translation quality on longer sentences (see Figure 6) and hurts even more when there is less training data [7].

In [5], a successful mechanism to overcome this problem was presented: a neural model of attention. In a sequence-to-sequence model with attention, one retains the outputs of all steps of the encoder and concatenates them to a *memory* tensor. At each step of the decoder, a probability distribution over this memory is computed and used to estimate a weighted average encoder representation to be used as input to the next decoder step. The decoder can hence focus on different parts of the encoder representation while producing tokens. Figure 1 illustrates a single step of this process.

The attention mechanism has proven useful well beyond the machine translation task. Image models can benefit from attention too; for instance, image captioning models can focus on the relevant parts of the image when describing it [8]; generative models for images yield especially good results with attention, as was demonstrated by the DRAW model [9], where the network focuses on a part of the

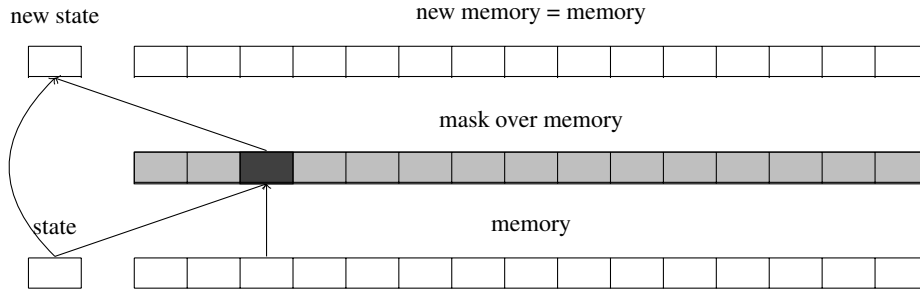

new state           new memory = memory

mask over memory

state          memory

Figure 1: Attention model. The state vector is used to compute a probability distribution over memory. Weighted average of memory elements, with focus on one of them, is used to compute the new state.

image to produce at a given time. Another interesting use-case for the attention mechanism is the Neural Turing Machine [10], which can learn basic algorithms and generalize beyond the length of the training instances.

While the attention mechanism is very successful, one important limitation is built into its definition. Since the attention mask is computed using a Softmax, it by definition tries to focus on a *single* element of the memory it is attending to. In the extreme case, also known as *hard attention* [8], one of the memory elements is selected and the selection is trained using the REINFORCE algorithm (since this is not differentiable) [11]. It is easy to demonstrate that this restriction can make some tasks almost unlearnable for an attention model. For example, consider the task of adding two decimal numbers, presented one after another like this:

| Input | 1 | 2 | 5 | 0 | + | 2 | 3 | 1 | 5 |
|---|---|---|---|---|---|---|---|---|---|
| Output | 3 | 5 | 6 | 5 | | | | | |

A recurrent neural network can have the carry-over in its state and could learn to shift its attention to subsequent digits. But that is only possible if there are *two* attention heads, attending to the first and to the second number. If only a single attention mechanism is present, the model will have a hard time learning this task and will not generalize properly, as was demonstrated in [12, 13].

A solution to this problem, already proposed in the recent literature (for instance, the Neural GPU from [12]), is to allow the model to access and change all its memory at each decoding step. We will call this mechanism an *active memory*. While it might seem more expensive than attention models, it is actually not, since the attention mechanism needs to compute an attention score for all its memory as well in order to focus on the most appropriate part. The approximate complexity of an attention mechanism is therefore the same as the complexity of the active memory. In practice, we get step-times around 1.7 second for an active memory model, the Extended Neural GPU introduced below, and 1.2 second for a comparable model with an attention mechanism. But active memory can potentially make parallel computations on the whole memory, as depicted in Figure 2.

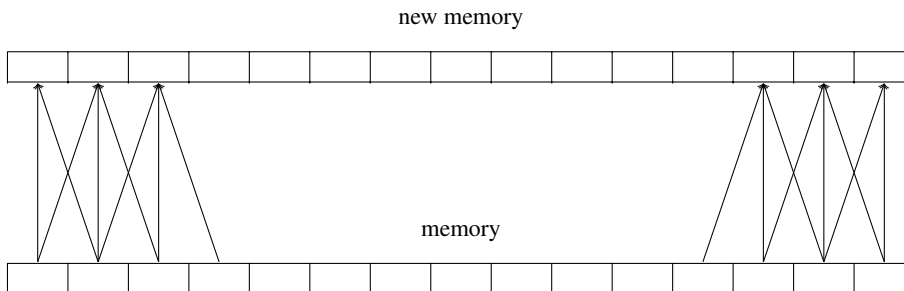

new memory

memory

Figure 2: Active memory model. The whole memory takes part in the computation at every step. Each element of memory is active and changes in a uniform way, e.g., using a convolution.

Active memory is a natural choice for image models as they usually operate on a canvas. And indeed, recent works have shown that actively updating the canvas that will be used to produce the final results can be beneficial. Residual networks [14], the currently best performing model on the ImageNet task, falls into this category. In [15] it was shown that the weights of different layers of a residual network can be tied (so it becomes recurrent), without degrading performance. Other models that operate on the whole canvas at each step were presented in [16, 17]. Both of these models are generative and show very good performance, yielding better results than the original DRAW model. Thus, the active memory approach seems to be a better choice for image models.

But what about non-image models? The Neural GPUs [12] demonstrated that active memory yields superior results on algorithmic tasks. But can it be applied to real-world problems? In particular, the original attention model brought a great success to natural language processing, esp. to neural machine translation. Can active memory be applied to this task on a large scale?

We answer this question positively, by presenting an extension of the Neural GPU model that yields good results for neural machine translation. This model allows us to investigate in depth a number of questions about the relationship between attention and active memory. We clarify why the previous active memory model did not succeed on machine translation by showing how it is related to the inherent dependencies in the target distributions, and we study a few variants of the model that show how a recurrent structure on the output side is necessary to obtain good results.

## 2   Active Memory Models

In the previous section, we used the term *active memory* broadly, referring to any model where every part of the memory undergoes active change at every step. This is in contrast to attention models where only a small part of the memory changes at every step, or where the memory remains constant.

The exact implementation of an active change of the memory might vary from model to model. In the present paper, we will focus on the most common ways this change is implemented that all rely on the *convolution* operator.

The convolution acts on a kernel bank and a 3-dimensional tensor. Our kernel banks are 4-dimensional tensors of shape $[k_w, k_h, m, m]$, i.e., they contain $k_w \cdot k_h \cdot m^2$ parameters, where $k_w$ and $k_h$ are kernel width and height. A kernel bank $U$ can be convolved with a 3-dimensional tensor $s$ of shape $[w, h, m]$ which results in the tensor $U * s$ of the same shape as $s$ defined by:

$$U * s[x, y, i] \;\; = \sum_{u = \lfloor -k_w/2 \rfloor}^{\lfloor k_w/2 \rfloor} \sum_{v = \lfloor -k_h/2 \rfloor}^{\lfloor k_h/2 \rfloor} \sum_{c=1}^{m} s[x + u, y + v, c] \cdot U[u, v, c, i].$$

In the equation above the index $x + u$ might sometimes be negative or larger than the size of $s$, and in such cases we assume the value is $0$. This corresponds to the standard convolution operator used in many deep learning toolkits, with zero padding on both sides and stride 1. Using the standard operator has the advantage that it is heavily optimized and can directly benefit from any new work (e.g., [18]) on optimizing convolutions.

Given a memory tensor $s$, an active memory model will produce the next memory $s'$ by using a number of convolutions on $s$ and combining them. In the most basic setting, a *residual* active memory model will be defined as:

$$s' = s + U * s,$$

i.e., it will only add to an already existing state.

While residual models have been successful in image analysis [14] and generation [16], they might suffer from the vanishing gradient problem in the same way as recurrent neural networks do. Therefore, in the same spirit as LSTM gates [3] and GRU gates [19] improve over pure RNNs, one can introduce convolutional LSTM and GRU operators. Let us focus on the convolutional GRU, which we define in the same way as in [12], namely:

$$\begin{aligned} \text{CGRU}(s) \;\; &= \;\; u \odot s + (1 - u) \odot \tanh(U * (r \odot s) + B), \quad \text{where} \\ u &= \sigma(U' * s + B') \quad \text{and} \quad r = \sigma(U'' * s + B''). \end{aligned} \tag{1}$$

As a baseline for our investigation of active memory models, we will use the Neural GPU model from [12], depicted in Figure 3, and defined as follows. The given sequence $i = (i_1, \ldots, i_n)$ of $n$ discrete

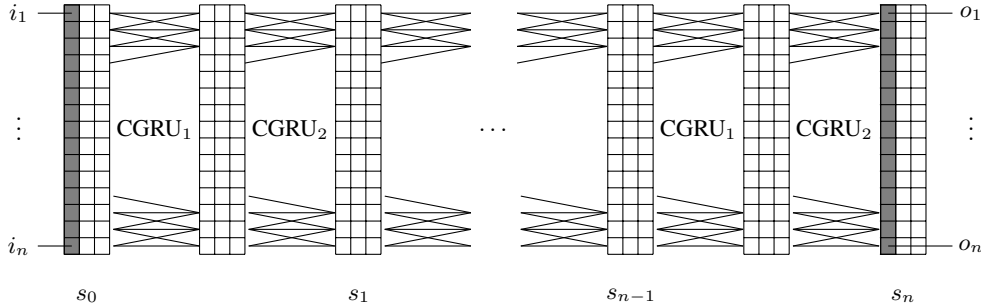

Figure 3: Neural GPU with 2 layers and width $w = 3$ unfolded in time.

symbols from $\{0, \dots, I\}$ is first embedded into the tensor $s_0$ by concatenating the vectors obtained from an embedding lookup of the input symbols into its first column. More precisely, we create the starting tensor $s_0$ of shape $[w, n, m]$ by using an embedding matrix $E$ of shape $[I, m]$ and setting $s_0[0, k, :] = E[i_k]$ (in python notation) for all $k = 1 \dots n$ (here $i_1, \dots, i_n$ is the input). All other elements of $s_0$ are set to 0. Then, we apply $l$ different CGRU gates in turn for $n$ steps to produce the final tensor $s_{\text{fin}}$:

$$s_{t+1} = \text{CGRU}_l(\text{CGRU}_{l-1} \dots \text{CGRU}_1(s_t) \dots) \quad \text{and} \quad s_{\text{fin}} = s_n.$$

The result of a Neural GPU is produced by multiplying each item in the first column of $s_{\text{fin}}$ by an output matrix $O$ to obtain the logits $l_k = O s_{\text{fin}}[0, k, :]$ and then selecting the largest one: $o_k = \text{argmax}(l_k)$. During training we use the standard loss function, i.e., we compute a Softmax over the logits $l_k$ and use the negative log probability of the target as the loss.

## 2.1 The Markovian Neural GPU

The baseline Neural GPU model yields very poor results on neural machine translation: its per-word perplexity on WMT[1] does not go below 30 (good models on this task go below 4), and its BLEU scores are also very bad (below 5, while good models are higher than 20). Which part of the model is responsible for such bad results?

It turns out that the main culprit is the output generator. As one can see in Figure 3 above, every output symbol is generated independently of all other output symbols, conditionally only on the state $s_{\text{fin}}$. This is fine for learning purely deterministic functions, like the toy tasks the Neural GPU was designed for. But it does not work for harder real-world problems, where there could be multiple possible outputs for each input.

The most basic way to mitigate this problem is to make every output symbol depend on the previous output. This only changes the output generation, not the state, so the definition of the model is the same as above until $s_{\text{fin}}$. The result is then obtained by multiplying by an output matrix $O$ each item from the first column of $s_{\text{fin}}$ concatenated with the embedding of the previous output generated by another embedding matrix $E'$:

$$l_k = O \, \text{concat}(s_{\text{fin}}[0, k, :], E' o_{k-1}).$$

For $k = 0$ we use a special symbol $o_{k-1} = \texttt{GO}$ and, to get the output, we select $o_k = \text{argmax}(l_k)$. During training we use the standard loss function, i.e., we compute a Softmax over the logits $l_k$ and use the negative log probability of the target as the loss. Also, as is standard in recurrent networks [4], we use teacher forcing, i.e., during training we provide the true output label as $o_{k-1}$ instead of using the previous output generated by the model. This means that the loss incurred from generating $o_k$ does not directly influence the value of $o_{k-1}$. We depict this model in Figure 4.

## 2.2 The Extended Neural GPU

The Markovian Neural GPU yields much better results on neural machine translation than the baseline model: its per-word perplexity reaches about 12 and its BLEU scores improve a bit. But these results are still far from those achieved by models with attention.

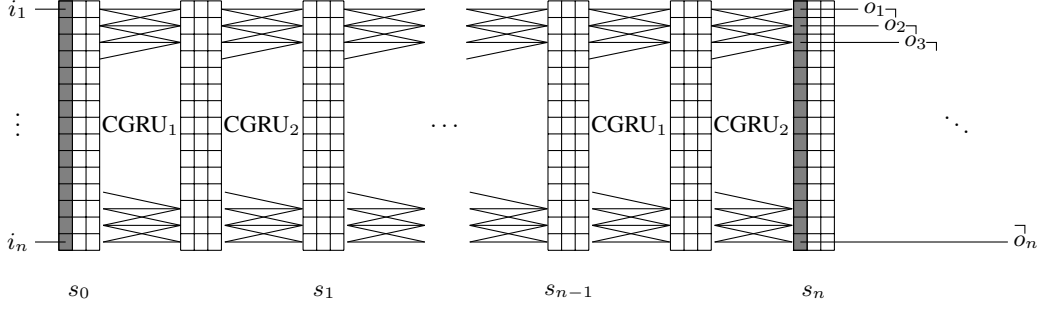

Figure 4: Markovian Neural GPU. Each output $o_k$ is conditionally dependent on the final tensor $s_{\text{fin}} = s_n$ and the previous output symbol $o_{k-1}$.

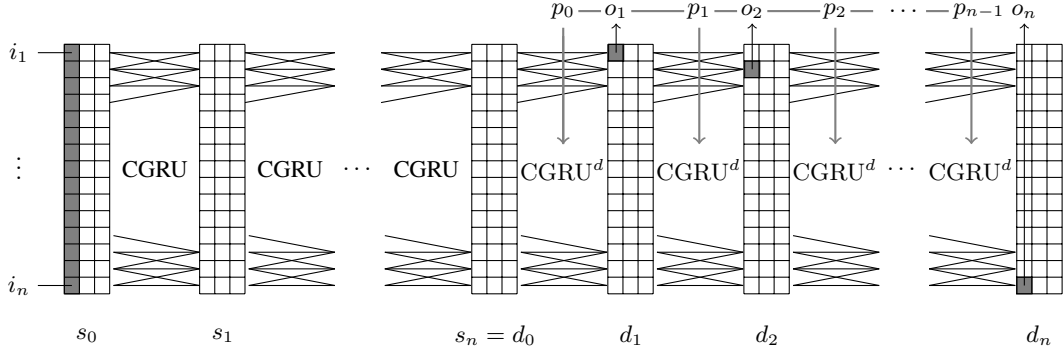

Figure 5: Extended Neural GPU with active memory decoder. See the text below for definition.

Could it be that the Markovian dependence of the outputs is too weak for this problem, that a full recurrent dependence of the state is needed for good performance? We test this by extending the baseline model with an *active memory decoder*, as depicted in Figure 5.

The definition of the Extended Neural GPU follows the baseline model until $s_{\text{fin}} = s_n$. We consider $s_n$ as the starting point for the active memory decoder, i.e., we set $d_0 = s_n$. In the active memory decoder we will also use a separate *output tape tensor* $p$ of the same shape as $d_0$, i.e., $p$ is of shape $[w, n, m]$. We start with $p_0$ set to all $0$ and define the decoder states by

$$d_{t+1} = \text{CGRU}_l^d(\text{CGRU}_{l-1}^d(\dots \text{CGRU}_1^d(d_t, p_t)\dots, p_t), p_t),$$

where $\text{CGRU}^d$ is defined just like CGRU in Equation (1) but with additional input as highlighted below in bold:

$$\begin{aligned}\text{CGRU}^d(s, p) &= u \odot s + (1 - u) \odot \tanh(U * (r \odot s) + \boldsymbol{W} * \boldsymbol{p} + B), \quad \text{where} \\ u &= \sigma(U' * s + \boldsymbol{W'} * \boldsymbol{p} + B') \quad \text{and} \quad r = \sigma(U'' * s + \boldsymbol{W''} * \boldsymbol{p} + B'').\end{aligned} \tag{2}$$

We generate the $k$-th output by multiplying the $k$-th vector in the first column of $d_k$ by the output matrix $O$, i.e., $l_k = O\, d_k[0, k, :]$. We then select $o_k = \text{argmax}(l_k)$. The symbol $o_k$ is then embedded back into a dense representation using another embedding matrix $E'$ and we put it into the $k$-th place on the output tape $p$, i.e., we define

$$p_{k+1} = p_k \quad \text{with} \quad p_k[0, k, :] \leftarrow E' o_k.$$

In this way, we accumulate (embedded) outputs step-by-step on the output tape $p$. Each step $p_t$ has access to all outputs produced in all steps before $t$.

Again, it is important to note that during training we use teacher forcing, i.e., we provide the true output labels for $o_k$ instead of using the outputs generated by the model.

### 2.3 Related Models

A convolutional architecture has already been used to obtain good results in word-level neural machine translation in [20] and more recently in [21]. These model use a standard RNN on top of the convolution to generate the output and avoid the output dependence problem in this way. But the state of this RNN has a fixed size, and in the first one the sentence representation generated by the convolutional network is also a fixed-size vector. Therefore, while superficially similar to active memory, these models are more similar to fixed-size memory models. The first one suffers from all the limitations of sequence-to-sequence models without attention [4, 6] that we discussed before.

Another recently introduced model, the Grid LSTM [22], might look less related to active memory, as it does not use convolutions at all. But in fact it is to a large extend an active memory model – the memory is on the *diagonal* of the grid of the running LSTM cells. The Reencoder architecture for neural machine translation introduced in that paper is therefore related to the Extended Neural GPU. But it differs in a number of ways. For one, the input is provided step-wise, so the network cannot start processing the whole input in parallel, as in our model. The diagonal memory changes in size and the model is a 3-dimensional grid, which might not be necessary for language processing. The Reencoder also does not use convolutions and this is crucial for performance. The experiments from [22] are only performed on a very small dataset of 44K short sentences. This is almost 1000 times smaller than the dataset we are experimenting with and makes is unclear whether Grid LSTMs can be applied to large-scale real-world tasks.

In image processing, in addition to the captioning [8] and generative models [16, 17] that we mentioned before, there are several other active memory models. They use *convolutional LSTMs*, an architecture similar to CGRU, and have recently been used for weather prediction [23] and image compression [24], in both cases surpassing the state-of-the-art.

## 3 Experiments

Since all components of our models (defined above) are differentiable, we can train them using any stochastic gradient descent optimizer. For the results presented in this paper we used the Adam optimizer [25] with $\varepsilon = 10^{-4}$ and gradients norm clipped to 1. The number of layers was set to $l = 2$, the width of the state tensors was constant at $w = 4$, the number of maps was $m = 512$, and the convolution kernels width and height was always $k_w = k_h = 3$.[2]

As our main test, we train the models discussed above and a baseline attention model on the WMT'14 English-French translation task. This is the same task that was used to introduce attention [5], but – to avoid the problem with the UNK token – we spell-out each word that is not in the vocabulary. More precisely, we use a 32K vocabulary that includes all characters and the most common words, and every word that is not in the vocabulary is spelled-out letter-by-letter. We also include a special SPACE symbol, which is used to mark spaces between characters (we assume spaces between words). We train without any data filtering on the WMT'14 corpus and test on the WMT'14 test set (newstest'14).

As a baseline, we use a GRU model with attention that is almost identical to the original one from [5], except that it has 2 layers of GRU cells, each with 1024 units. Tokens from the vocabulary are embedded into vectors of size 512, and attention is put on the top layer. This model is identical as the one in [7], except that is uses GRU cells instead of LSTM cells. It has about 120M parameters, while our Extended Neural GPU model has about 110M parameters. Better results have been reported on this task with attention models with more parameters, but we aim at a baseline similar in size to the active memory model we are using.

When decoding from the Extendend Neural GPU model, one has to provide the expected size of the output, as it determines the size of the memory. We test all sizes between input size and double the input size using a greedy decoder and pick the result with smallest log-perplexity (highest likelihood). This is expensive, so we only use a very basic beam-search with beam of size 2 and no length normalization. It is possible to reduce the cost by predicting the output length: we tried a basic estimator based just on input sentence length and it decreased the BLEU score by 0.3. Better training and decoding could remove the need to predict output length, but we leave this for future work.

| Model | Perplexity (log) | BLEU |
|---|---|---|
| Neural GPU | 30.1 (3.5) | < 5 |
| Markovian Neural GPU | 11.8 (2.5) | < 5 |
| Extended Neural GPU | 3.3 (1.19) | **29.6** |
| GRU+Attention | 3.4 (1.22) | 26.4 |

Table 1: Results on the WMT English->French translation task. We provide the average per-word perplexity (and its logarithm in parenthesis) and the BLEU score. Perplexity is computed on the test set with the ground truth provided, so it do not depend on the decoder.

For the baseline model, we use a full beam-search decoder with beam of size 12, length normalization and an attention coverage penalty in the decoder. This is a basic penalty that pushes the decoder to attend to all words in the source sentence. We experimented with more elaborate methods following [27] but it did not improve our results. The parameters for length normalization and coverage penalty are tuned on the development set (newstest'13). The final BLEU scores and per-word perplexities for these different models are presented in Table 1. Worse models have higher variance of their BLEU scores, so we only write $< 5$ for these models.

One can see from Table 1 that an active memory model can indeed match an attention model on the machine translation task, even with slightly fewer parameters. It is interesting to note that the active memory model does not need the length normalization that is necessary for the attention model (esp. when rare words are spelled). We conjecture that active memory inherently generalizes better from shorter examples and makes decoding easier, a welcome news, since tuning decoders is a large problem in sequence-to-sequence models.

In addition to the summary results from Table 1, we analyzed the performance of the models on sentences of different lengths. This was the key problem solved by the attention mechanism, so it is worth asking if active memory solves it as well. In Figure 6 we plot the BLEU scores on the test set for sentences in each length bucket, bucketing by 10, i.e., for lengths $(0, 10], (10, 20]$ and so on. We plot the curves for the Extended Neural GPU model, the long baseline GRU model with attention, and – for comparison – we add the numbers for a non-attention model from Figure 2 of [5]. (Note that these numbers are for a model that uses different tokenization, so they are not fully comparable, but still provide a context.)

As can be seen, our active memory model is less sensitive to sentence length than the attention baseline. It indeed solves the problem that the attention mechanism was designed to solve.

**Parsing.** In addition to the main large-scale translation task, we tested the Extended Neural GPU on English constituency parsing, the same task as in [7]. We only used the standard WSJ dataset for training. It is small by neural network standards, as it contains only 40K sentences. We trained the Extended Neural GPU with the same settings as above, only with $m = 256$ (instead of $m = 512$) and dropout of $30\%$ in each step. During decoding, we selected well-bracketed outputs with the right number of POS-tags from all lengths considered. Evaluated with the standard EVALB tool on the standard WSJ 23 test set, we got $85.1$ F1 score. This is lower than $88.3$ reported in [7], but we didn't use any of their optimizations (no early stopping, no POS-tag substitution, no special tuning). Since a pure sequence-to-sequence model has F1 score well below 70, this shows that the Extended Neural GPU is versatile and can learn and generalize well even on small data-sets.

## 4   Discussion

To better understand the main shortcoming of previous active memory models, let us look at the average log-perplexities of different attention models in Table 1. A pure Neural GPU model yields $3.5$, a Markovian one yields $2.5$, and only a model with full dependence, trained with teacher forcing, achieves $1.3$. The recurrent dependence in generating the output distribution turns out to be the key to achieving good performance.

We find it illuminating that the issue of dependencies in the output distribution can be disentangled from the particularities of the model or model class. In earlier works, such dependence (and training with teacher forcing) was always used in LSTM and GRU models, but very rarely in other kinds

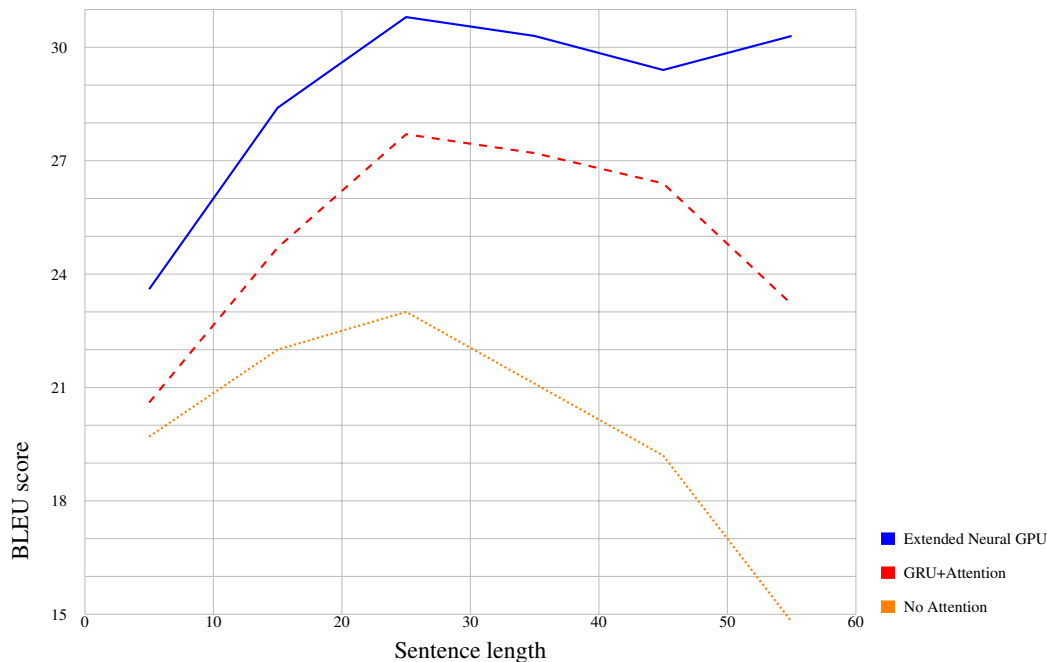

Figure 6: BLEU score (the higher the better) vs source sentence length.

models. We show that it can be beneficial to consider this issue separately from the model architecture. It allows us to create the Extended Neural GPU and this way of thinking might also prove fruitful for other classes of models.

When the issue of recurrent output dependencies is addressed, as we do in the Extended Neural GPU, an active memory model can indeed match or exceed attention models on a large-scale real-world task. Does this mean we can always replace attention by active memory?

The answer could be **yes** for the case of soft attention. Its cost is approximately the same as active memory, it performs much worse on some tasks like learning algorithms, and – with the introduction of the Extended Neural GPU – we do not know of a task where it performs clearly better.

Still, an attention mask is a very natural concept, and it is probable that some tasks can benefit from a selector that focuses on single items by definition. This is especially obvious for hard attention: it can be used over large memories with potentially much less computational cost than an active memory, so it might be indispensable for devising long-term memory mechanisms. Luckily, active memory and attention are not exclusive, and we look forward to investigating models that combine these mechanisms.

## Footnotes

[1]See Section 3 for more details on the experimental setting.

[2]Our model was implemented using TensorFlow [26]. Its code is available as open-source at `https://github.com/tensorflow/models/tree/master/neural_gpu/`.

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
