[Reviews · NeurIPS 2016]

Reviewer 1

Summary

The authors propose to replace the notion of 'attention' in neural architectures with the notion of 'active memory' where rather than focusing on a single part of the memory one would operate on the whole of it in parallel.

Qualitative Assessment

The author build on the Neural GPU architecture to make it applicable to large scale real world tasks. They analyze the shortcoming at each phase, and propose a series of improvements. The findings are of interest for the community. Particularly interesting is the study of performance dependency on the input complexity/length where they show to achieve higher robustness and hence they conclude that this can be an indication that active memory approaches can replace active memory. A more detailed study on an artificial task whose complexity can be fully controlled would improve the paper as it would show the dependency of the network complexity wrt the task complexity.

Confidence in this Review

2-Confident (read it all; understood it all reasonably well)


Reviewer 2

Summary

This paper introduces an extension to neural GPUs for machine translation. I found the experimental analysis section lacking in both comparisons to state of the art MT techniques as well as thoroughly evaluating the proposed method.

Qualitative Assessment

I found the model section to be very hard to follow. I would like to see a more thorough evaluation of the parameters and design decisions of this model.

Confidence in this Review

2-Confident (read it all; understood it all reasonably well)


Reviewer 3

Summary

This paper proposes active memory, which is a memory mechanism that operates all the part in parallel. The active memory was compared to attention mechanism and it is shown that the active memory is more effective for long sentence translation than the attention mechanism in English-French translation.

Qualitative Assessment

- Figure 6 should be displayed well without color. - Is the BLEU score of the extended neural GPU statistically significant improvement against the GRU+attention model? - One language pair of translation is not strong experimental results. You need more experiments to support your claim. - Conclusion section should exist.

Confidence in this Review

1-Less confident (might not have understood significant parts)


Reviewer 4

Summary

This paper proposes two new models for modeling sequential data in the sequence-to-sequence framework. The first is called the Markovian Neural GPU and the second is called the Extended Neural GPU. Both models are extensions of the Neural GPU model (Kaiser and Sutskever, 2016), but unlike the Neural GPU, the proposed models do not model the outputs independently but instead connect the output token distributions recursively. The paper provides empirical evidence on a machine translation task showing that the two proposed models perform better than the Neural GPU model and that the Extended Neural GPU performs on par with a GRU-based encoder-decoder model with attention.

Qualitative Assessment

The contributions of this paper comes from the proposed Extended Neural GPU model and from the empirical results demonstrating that it performs on par with an attention mechanism. The contribution of extending the model by modeling the output sequence dependencies has not been applied to the Neural GPU specifically, but it is well-established in the literature (e.g. LSTMs and GRU RNN decoders for language modeling, machine translation, image captioning etc). On the other hand, the experimental contribution of making the Extended Neural GPU model work effectively on a machine translation task is useful, and it is especially interesting to see that such an architecture may yield the same advantages as an attention mechanism,. The need for a variable-sized memory is partly supported by (Cho et al., 2014), who demonstrate that the performance of an encoder-decoder translation model, where the encoder is a convolutional neural network, also degrades with sentence length. This adds evidence to the paper's argument that the memory should not be restricted to a fixed-sized vector, but instead allowed to grow with the input sequence length. If this paper is to be published, it needs to at least address the following issues: 1. The output tape tensor "p" should be defined formally. It is not clear to me how it is computed based on the description on lines 144-147 alone. It is also not clear to me, how the previous decoder states, e.g. d_t, affect future decoder states, e.g. d_{t+1}. This should be clarified. 2. On line 207, the paper states that the greedy decoding strategy is comparable to the decoding strategy of the Extended Neural GPU. In my opinion this is incorrect, since the Extended Neural GPU runs a separate greedy decoder for every output size in the interval [input size, 2 x input size]. In this case, the later is much better and more expensive than the former and therefore the GRU+Attention (short) is not really comparable to the Extended Neural GPU model. In addition to these two points, it would also be good if the final paper includes results for the Extended Neural GPU model on the WMT' 14 task with long sentences, as well as a plot showing performance of the different models w.r.t. beam size in order to demonstrate that the Extended Neural GPU is not benefiting significantly from the alternative decoding procedure. Other comments: - Line 106: is {0, ..., I} the vocabulary with size I? If so, it would be more clear to explicitly define the vocabulary with its own symbol. - The paper should include an appropriate reference to the WMT' 14 task. - Lines 239-244: The statements made in this paragraph are already well established. The paragraph should be shortened to a single sentence. - Reference section should be improved. ------------------------------------ UPDATE: Thanks to the authors for their rebuttal. The clarification on beam search helped, and so did the improved results.

Confidence in this Review

2-Confident (read it all; understood it all reasonably well)


Reviewer 5

Summary

This paper proposed extended model of active memory that matches existing attention models on neural machine translation. Also, the paper try to investigate this model and explain why previous active memory models did not succeed.

Qualitative Assessment

It would be informative to compare the training time & test time between proposed model and attention model, because the active memory model seems to have more computation. In addition, Figure 6 should compare much longer source sentence length, because the long baseline GRU model with attention is trained including all sentences up to 128 tokens. The paper also didn't compare their results with previous papers.

Confidence in this Review

2-Confident (read it all; understood it all reasonably well)


Reviewer 6

Summary

This paper is built on the top of Neural GPU model by making changes to the decoder to handle sequence generation tasks like machine translation better. While the original Neural GPU model was applied for algorithmic tasks, author try to use this model for machine translation task. Results show that proposed active memory model is comparable to Attention model for NMT.

Qualitative Assessment

After Rebuttal: I have read the rebuttal and I see that authors have addressed most of my concerns. I am changing my rating for impact from 2 to 3. Before Rebuttal: This paper proposes 2 extensions to Neural GPU model. Markovian Neural GPU is an obvious extension of Neural GPU for sequence generation. Extended Neural GPU is a non-trivial extension. Firstly the claim (in abstract and everywhere) that active memory has not improved over attention for NLP tasks is wrong. Dynamic Memory Networks and End-to-end memory networks have been used for comprehension based question answering task. For machine translation, Meng et al., 2016 (http://arxiv.org/pdf/1506.06442v4.pdf) proposes a deep memory based architecture which is an active memory model and it does perform better than the attention based NMT model. In fact their model is very similar to Neural GPU, except for the fact that they use a different transformation instead of convolution. Input structure as described in line 106-110 is not very clear. If I understand correctly, authors use w*n*m matrix as input where only w[0] is filled with n m-dimensional word vectors and w[k] for k > 0 are all set to 0? Also if number of words are less than n, then remaining part of the w[0] matrix are also set to 0? Please clarify. From lines 39-41 I understand that authors consider NTM as a complex attention model than an active memory model? If so, I disagree with that point of view. Experiments: 1. I understand that the authors want to compare their model with attention based model without beam search. But I do not understand the reason for choosing two attention models – one trained with sentences upto 64 tokens and another one with sentences upto 128 tokens. Why is Extended Neural GPU trained with sentences upto 64 tokens only? Specifically, the comparison is incomplete without the following rows: a. Attention model trained with sentences upto 64 + beam search b. Attention model trained with sentence upto 128 tokens + greedy decoder. c. Neural GPU models trained with sentences upto 128 tokens. d. Meng et al., 2016 2. Authors should also highlight the limitations of the proposed model clearly. a. Is it not possible to train Neural GPU with 128 tokens? Or the results are worse? In either case, this is a limitation. b. Neural GPU models expect the output sentence size. In lines 203-208, authors say that they consider all sizes between input size and double of it and pick the result with smallest perplexity. This is very inefficient when compared to simple decoding done with LSTM decoder. c. I see that it is very difficult to do beam search with Neural GPU, because of the proposed complex decoder. This is another limitation. 3. Just based on Figure 6 (which is approximate), authors cannot make the claim that less sensitive to sentence length than the attention baseline. Because other 2 models where trained with UNK token and few other variations. Overall, this paper proposes an interesting extension to Neural GPU, but the application is not very convincing. It is not very clear, what is the gain in using a complex architecture like this when compared to the simple attention model if the results are only comparable and not better. I would like to see a comparison with Meng et al., 2016 to see if this complex architecture really helps to improve the performance.

Confidence in this Review

3-Expert (read the paper in detail, know the area, quite certain of my opinion)